# Anaerobic consortia of fungi and sulfate reducing bacteria in deep granite fractures

Henrik Drake [1], Magnus Ivarsson[2], Stefan Bengtson [2], Christine Heim [3], Sandra Siljeström[4], Martin J. Whitehouse[5], Curt Broman[6], Veneta Belivanova[2] & Mats E. Åström[1]

The deep biosphere is one of the least understood ecosystems on Earth. Although most microbiological studies in this system have focused on prokaryotes and neglected microeukaryotes, recent discoveries have revealed existence of fossil and active fungi in marine sediments and sub-seafloor basalts, with proposed importance for the subsurface energy cycle. However, studies of fungi in deep continental crystalline rocks are surprisingly few. Consequently, the characteristics and processes of fungi and fungus-prokaryote interactions in this vast environment remain enigmatic. Here we report the first findings of partly organically preserved and partly mineralized fungi at great depth in fractured crystalline rock (−740 m). Based on environmental parameters and mineralogy the fungi are interpreted as anaerobic. Synchrotron-based techniques and stable isotope microanalysis confirm a coupling between the fungi and sulfate reducing bacteria. The cryptoendolithic fungi have significantly weathered neighboring zeolite crystals and thus have implications for storage of toxic wastes using zeolite barriers.

[1] Department of Biology and Environmental Science, Linnæus University, Kalmar 39182, Sweden. [2] Department of Palaeobiology and Nordic Center for Earth Evolution (NordCEE), Swedish Museum of Natural History, P.O. Box 50 007, Stockholm 10405, Sweden. [3] Geoscience Centre Göttingen of the Georg-August University (Department of Geobiology), Goldschmidtstr. 3, Göttingen 37077, Germany. [4] Department of Surfaces, Chemistry and Materials, SP Technical Research Institute of Sweden, P.O. Box 857, Borås 50115, Sweden. [5] Department of Geosciences and Nordic Center for Earth Evolution (NordCEE), Swedish Museum of Natural History, P.O. Box 50007, Stockholm 10405, Sweden. [6] Department of Geological Sciences, Stockholm University, Stockholm 106 91, Sweden. Correspondence and requests for materials should be addressed to H.D. (email: henrik.drake@lnu.se)

The deep subsurface biosphere comprises microorganisms several kilometers below the surface[1]. Microbiological investigations have revealed active deep ecosystems in marine sediments[2], deep-sea hydrothermal vents[3], sub-seafloor igneous rocks[4], and terrestrial sedimentary[5] and crystalline rocks[6]. Owing to its recent discovery and the difficulties in accessing samples, the deep biosphere is among the least understood ecosystems on Earth. Although processes are relatively slow because of the low energy supply[7] the deep ecosystems are proposed to comprise a significant biomass and play an important role in the energy cycling of the Earth[8]. Estimates from the deep continental subsurface suggest that this environment accommodates a significant part of the Earth's biomass (up to 19%)[8]. Until recently, the majority of the microbiological investigations in the deep biosphere have been focused on prokaryotes, and the potential presence of eukaryotes such as fungi has been largely neglected[9, 10]. Recently, fungi have been found to exist and have been isolated from various deep marine settings[10–14] including sub-seafloor basalt[9, 15–17], suggesting that fungi play a major role in the element and energy cycling[9]. In continental deep settings studies involving fungi are surprisingly rare. Reitner et al.[18] described fossilized putative fungal hyphae in the Triberg granite, Germany, and Ivarsson et al.[19] described fossilized mycelia from the Lockne impact structure, Sweden, of Ordovician age. Ekendahl et al.[20] isolated a few strains of yeast fungi from fluids at Äspö, Sweden, and Sohlberg et al.[21] examined the total fungal diversity in anaerobic bedrock fractures at Olkiluoto, Finland, and found the diversity to be higher than expected consisting of most major fungal phyla, some minor phyla and even novel species. At great depth in continental granite aquifer systems anoxic conditions prevail, and the fungi living there are considered anaerobic[21]. Anaerobic fungi are so far poorly understood in an environmental context, reported only from a few anoxic settings[14, 22], but are most thoroughly described from rumina of ruminating herbivores[23, 24]. Because of the production of $H_2$ during their respiration, anaerobic fungi consort with $H_2$-dependent methanogenic and acetogenic archaea in the rumen, which enhances growth of both organisms. Potentially any $H_2$-dependent chemoautotrophic microorganism could be fuelled by anaerobic fungi in an anoxic environment[25] and it has been suggested that anaerobic fungi represent a neglected geobiological force in the subsurface ecosystem[9]. Direct evidence of such consortia in the subsurface remains to be confirmed. However, the fungi that form symbiotic relationships with acetogens and methanogens in the rumen have recently been described from marine sediments[26]. Even though the presence of fungi in the continental crystalline basement is confirmed by a few reports, there is a huge gap in knowledge compared to what is known about the prokaryotes, and there is an urgent need to investigate the abundance, diversity and ecological role of fungi in these deep environments.

Here we present an extensive study of previously unseen organically preserved and partly mineralized fungal hyphae, inferred as anaerobic fungi, from deep fractured continental crystalline rocks (740 m depth at the Laxemar site in Sweden). Utilization of state-of-the-art methodology including secondary ion mass spectrometry (SIMS), synchrotron radiation X-ray tomographic microscopy (SRXTM), and Time-of-Flight (ToF-) SIMS enables comprehensive new characterisation of the fungi, the cryptoendolithic behaviour of fungi, and the prokaryote-fungus interaction in this environment. This not only increases the understanding of the role fungi play in the energy cycling of the deep biosphere, but also has societal implication for long-term storage of toxic wastes.

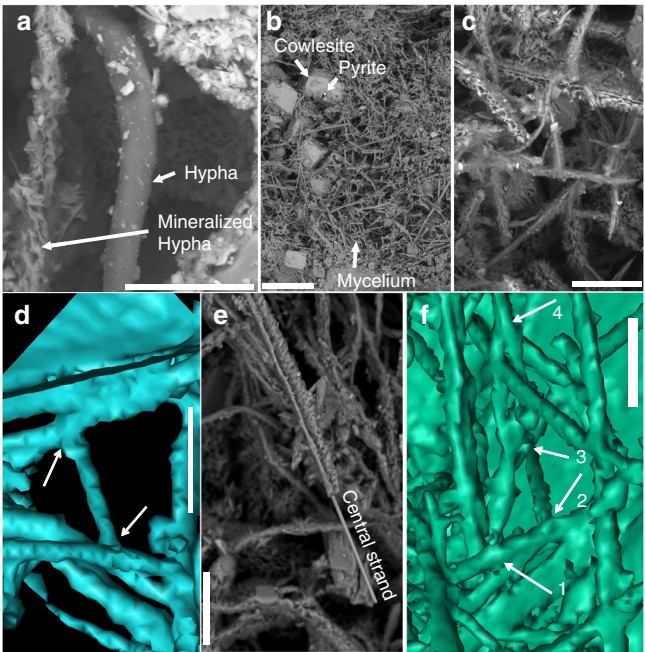

**Fig. 1** Hypha characteristics. Back-scattered Environmental Scanning Electron Microscopy ESEM-images (**a**–**c**, **e**) and Synchrotron Radiation X-ray Tomographic Microscopy (SRXTM) surface renderings (**d**, **f**). (**a**) Organically preserved hypha (10 μm in diameter) next to a mineralized hypha of the same diameter. (**b**) Mycelium like structures partly overgrowing cowlesite. (**c**) Branching and anastomosing mineralized hyphae. (**d**) Branching (T-junction) and anastomosing hyphae (indicated by *arrows*). (**e**) Mineralized central strand in hyphae. (**f**) 3D-projection of the hyphae on the fracture surface, showing 1. Y-junction. 2. T-junction + anastomos. 3. Y-junction + anastomos. 4. Y-junction. Scale marker bars are 40 μm (**a**), 500 μm (**b**), 50 μm (**c**), 30 μm (**d**), 50 μm (**e**) and 30 μm (**f**)

## Results

**Fossilized Microorganisms.** Mycelial-like networks consisting of long filamentous structures were discovered at 740 m depth in a drill core (KLX09) at the Laxemar site, Sweden, where the Swedish Nuclear Fuel and Waste Management Co. (SKB) recently carried out site investigations for a deep nuclear-waste repository. At this site evidence of both active[27, 28] and ancient prokaryotic activity, such as bacterial sulfate reduction[29] and anaerobic oxidation of methane[30], has been reported. The mycelial-like networks occur in a partly open fracture in a meter-wide quartz vein (Supplementary Fig. 1). The fracture walls are coated by euhedral zeolite, identified as cowlesite by Raman spectroscopic analysis, as well as calcite, pyrite and clay minerals. The filaments cover an area of about 25–30 cm² of the fracture walls in the drill core sample. The filaments are undulating and typically have a diameter of 5–20 μm (Figs 1a and 2), consistent throughout the filaments, while the length of the filaments extends up to several hundred micrometres in length. The filaments branch frequently, forming both Y- and T-junctions and occasional anastomoses between branches (Fig. 1b–d, f). The overall architecture of the networks is an intertwined, mycelium-like structure where the filaments are interconnected (Fig. 1c). The filaments are characterized by a central strand encircled by a thicker marginal wall. At places where filaments have been broken off, probably during sampling or preparation, the central strand is clearly visible in cross-section occupying about a third of the total filament diameter (Fig. 1e).

The filaments are either mineralized or carbonaceous in composition. Energy Dispersive X-ray Analysis (EDX) and

Raman spectroscopic analysis show that the mineralized parts of the fossilized filaments are dominated by Fe/Mg/Ca-rich clay minerals, together with some minor Fe-oxide. The central strand is composed of Fe-oxides and surrounded by the clay phase. EDX analyses indicate the presence of carbon throughout the filaments, even where these are mineralized.

Raman spectroscopic analysis on the mineralized filaments and the carbonaceous filaments shows characteristics of typical organic compounds with three clear peaks at 1300, 1440 and 1660 $cm^{-1}$, which may be attributed to $CH_2$ and $CH_3$ deformations and C=C stretching vibrations similar to what has been found in fatty acids[31]. The spectra further show several bands between 2720 and 3300 $cm^{-1}$ with the most intense at 2850 and 2930 $cm^{-1}$ assigned to $CH_2$ and $CH_3$ stretching vibrations.

There are detectable biomarkers in the carbonaceous filaments, but none of them are specific for prokaryotes or eukaryotes. The gas chromatography (GC)–MS analyses mainly detected fatty acids and aromatic substances like dimethylnaphthalenes, trimethylnaphthalenes, and methylbiphenyls (Fig. 3 and

Supplementary Table 1), which are characteristic for thermal maturation. The Time-of-Flight (ToF)-SIMS results of partly mineralized hyphae show possible association of sulfur (pyrite), $SO_4H^-$ and $PO_2$ with sugar fragments (potentially chitin-related fragment as the $C_3H_3O_2$ ion produces a strong signal in the chitin standard spectrum, but also in spectra of other types of sugar molecules), and fatty acids ($C_{16:0}$ and $C_{18:1}$ at $m/z$ 255.23 and 281.25, respectively, Supplementary Fig. 2). The mineralized filaments and carbonaceous filaments occur side by side as parts of the same mycelium but at various stages of fossilization (Fig. 1a). There are also previously unseen gradual transitions from carbonaceous parts to mineralized parts in single filaments (Fig. 4). In addition to the filaments, there is a carbonaceous/ partly mineralized biofilm, a few micrometres thick, that covers parts of the minerals and from which some filaments, especially the carbonaceous ones, originate (Fig. 2b).

Significant weathering has occurred at the contact between the mycelium-forming microorganism (including the biofilm) and the secondary mineralizations (zeolite and calcite, Fig. 2). Such weathering is absent in minerals not covered by the mycelium. The minerals have a rough and clearly etched surface when associated with the mycelium. Filaments creeping along a mineral surface leave an etched gorge-like channel in the surfaces (Fig. 2c, d). Filaments are also continually penetrating the minerals through micro-fractures as revealed by the SRXTM investigations (Fig. 2e).

**Minerals.** Pyrite crystals are widely distributed out in the fracture void (Fig. 5a), in particular in association with hyphae, and occur as two different generations. The older pyrite generation consists of cubic and octahedral crystals ranging between 30 and 70 μm (Fig. 6a), and the younger more fine-grained pyrite (2–25 μm) occurs partly as overgrowths on the older pyrite generation (Fig. 6a) but mainly as fine-grained cubic crystals on the mycelial-forming filaments (Figs 5b–e and 6b). Detailed micro-scale analyses ($n = 56$) of S-isotopes show that the older pyrite generation ranges in $\delta^{34}S$ from +12.3 to +30.4‰ V-CDT (denoting Vienna Canyon Diablo Troilite), with most values in a quite narrow span around +23 ± 4‰ (Fig. 6c). The younger pyrite generation has $\delta^{34}S$ values relatively evenly distributed from −53.3 to +0.3‰ (Figs 6c, d and Supplementary Table 2).

Detailed micro-scale isotope analyses were also carried out on calcite ($n = 47$ for C-isotopes and $n = 44$ for O-isotopes). The variability of $\delta^{13}C$ was 42.7‰ V-PDB, (denoting Vienna Pee Dee Belemnite) and of $\delta^{18}O$ 16.5‰ V-PDB (Fig. 7c; Supplementary Table 3). The calcite crystals show distinct growth zonation that represents at least three precipitation events (generations) with specific stable isotope compositions. Analytical transects within the crystals show that the oldest generation has highest $\delta^{13}C$ and lowest $\delta^{18}O$ (Fig. 7a, d, zoned crystal with growth direction from

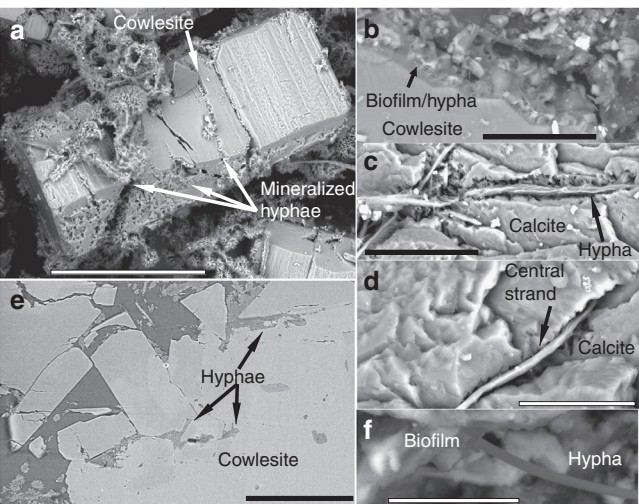

**Fig. 2** Cryptoendolithic features of hyphae and weathering of zeolite and calcite. Back-scattered Environmental Scanning Electron Microscopy (ESEM)-images (**a**–**d**, **f**) and Synchrotron Radiation X-ray Tomographic Microscopy (SRXTM)-section (**e**). (**a**) Heavily wheathered cowlesite (zeolite mineral) covered by hyphae (now mineralized). (**b**) Chemical weathering front in the contact between fungi and cowlesite. (**c**, **d**) Hyphae penetrating micro-fractures into calcite. Only the central strand of the hyphae is remaining in most parts. (**e**) SRXTM cross-section showing hyphae between cowlesite grains and within cracks in the minerals. (**f**) Biofilm on mineral surfaces with hypha extending out from the biofilm. Scale marker bars are 200 μm (**a**), 30 μm (**b**), 40 μm (**c**), 20 μm (**d**), 300 μm (**e**) and 10 μm (**f**)

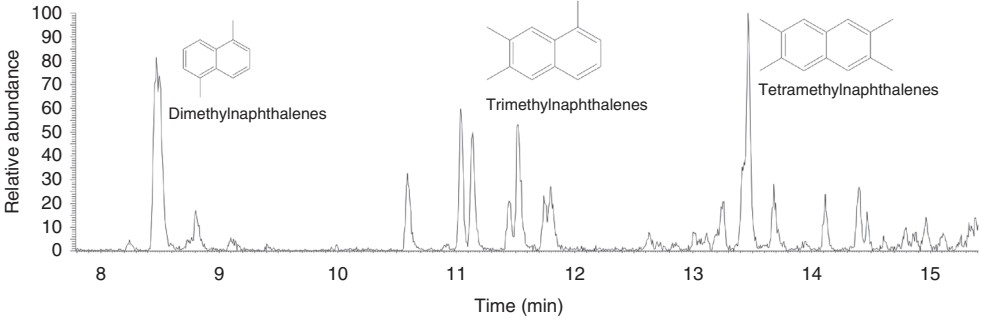

**Fig. 3** GC–MS data of aromatic fraction. Fragmentograms $m/z$ 156, $m/z$ 170 and $m/z$ 184: distributions of naphthalenes and its alkyl derivatives, from the fungi

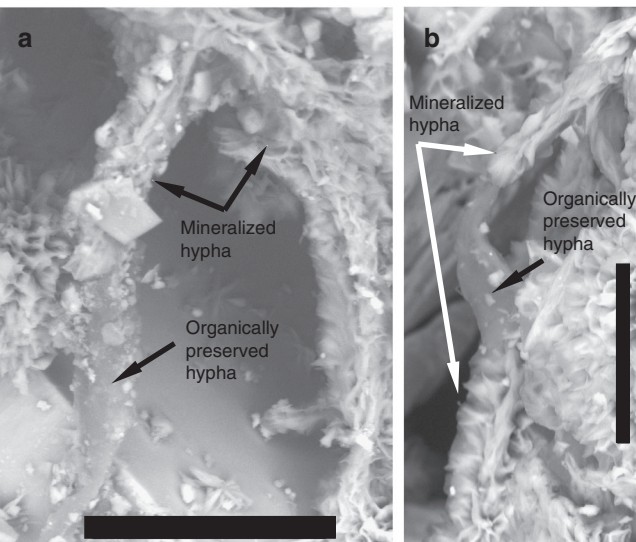

**Fig. 4** Gradual transition between organically preserved and mineralized hyphae. Back-scattered Environmental Scanning Electron Microscopy (ESEM)-images. Scale bars are 50 μm (**a**) and 30 μm (**b**)

left to right). This is followed by a generation strongly depleted in $^{13}C$ (−43 to −26‰) and with highest $\delta^{18}O$ (−11 to −4.4‰), whereas the youngest generation with $\delta^{13}C$ of −24 to −16‰ and $\delta^{18}O$ of −15 to −11‰ (Fig. 7b, e) is most affected by microorganism-related chemical weathering. Single phase liquid fluid inclusions in the calcite (Supplementary Fig. 3) indicate formation temperatures below 50 °C[32]. The inclusions show low eutectic melting temperature close to −50 °C, which points to a composition where $CaCl_2$ is the most abundant salt. Final ice melting temperatures in the range of −22.4 to −23.5 °C correspond to salinities of 21.7–22.2 wt.% $CaCl_2$.

## Discussion

The mycelium-like appearance, the diameter, and the anastomosing behavior of the filaments (Fig. 1) are all distinctive features of fungi. The presence of a mineralized central strand has been shown as a common feature among fossilized fungal hyphae[33–36]. Lining of mineral surfaces by a basal biofilm from which further hyphal growth emanates to form a mycelium is also typical for endolithic fungi[15, 19, 33, 34]. Except for fungi, actinobacteria and the stramenopile oomycetes are the only microorganisms forming mycelium-like networks of branching filaments. However, actinobacterial filaments never exceed 2 μm in diameter and anastomoses have not been confirmed[37, 38]. Thus, the diameter of the current filaments, together with the presence of anastomoses excludes an actinobacterial interpretation. Oomycete filaments have been reported to form anastomoses, but as a means of conjugation rather than as a structural feature[39]. Based on these morphological features we infer a fungal interpretation of the networks and suggest that they represent diagenetically mineralized fungal hyphae. As further support for this, the weathered mineral surfaces in contact with the fungi bear close resemblance to fungal induced weathering seen in many other minerals, owing to production of organic acids[40]. Active boring seen among fungi in sub-seafloor basalts is not observed in the sample of the present study. However, the fungal hyphae typically exploit microfractures in the minerals for penetration.

The partly mineralized nature is in itself a rare finding, which has previously only been observed in the laboratory[41, 42]. Usually, subsurface fungi are completely fossilized to clay minerals and Fe-oxides with rare carbonaceous elements and no

biomarkers at all[33, 34], except rare chitin observations[16]. Our findings give insights into the fossilization process of fungi through a transition from maturation of the organic matter to a carbonaceous material, before being finally mineralized by clays and Fe-oxides. Based on our observations the mineralization starts from the centre of the hypha with a fully mineralized Fe-oxide dominated central strand and clay-mineral dominated margins. The negatively charged carbonaceous material attracts the positively charged Si, Al, Mg, Fe (and minor Na, and Ca) cations of the clays. Initial adsorption of cations on the carbonaceous hyphae sparks subsequent adsorption and clay mineralization[34, 43]. The end result of complete mineralization by clays and Fe-oxides is in agreement with previous observations of deep fossilized fungi and supports what seems to be an overall characteristic pattern of fungal fossilization in deep ecosystems.

Fungi are heterotrophs and need access to carbohydrates like mono- or polysaccharides for their metabolism. In the deep oligotrophic granite environment, the most likely source of carbohydrates is living or dead bacterial biofilms[44]. The large $^{13}C$-depletions of the calcite, resulting in $\delta^{13}C$ values as low as −43‰ (Fig. 7c), point to oxidation of methane in the fracture system[45], as described previously for this setting[30]. The youngest generation of calcite shows $\delta^{13}C$ values suggesting microbial degradation of organic C. Remnant biofilms of metanotrophs, as well as sulfate-reducing bacteria (SRB) that evidently have occupied the fracture at several occasions, may have acted as nutrients for the fungi and triggered the fungal colonization of the system.

The fungi cannot be taxonomically classified in detail because of the fossilization and lack of morphological features like septa, but they are considered to have grown in an anaerobic environment. A major support for this is that the current groundwater in the fracture network turns anoxic in the upper tens of meters[46] and below that depth remains strictly reducing, with Eh values in the range of −300 to −200 mV[47]. Based on these features, it seems highly unlikely that the conditions in the paleo-groundwater would have been oxidising or suboxic at a depth as great as 740 m. There are several additional lines of support for prolonged anoxic conditions. First, there is pyrite in relation to the hyphae; second, oxidation-related alteration features are not detected in any of the pyrite generations in the fractures, in contrast to pyrite at shallower depth where waters with dissolved oxygen have infiltrated[46]; third, Ce(IV) and positive Ce anomalies, which are indicative of oxidising conditions, are frequent in fracture coatings in the upper 10 m of the bedrock but absent below that depth[48]; and, fourth, abundant signs of anaerobic oxidation of methane ($^{13}C$-depleted calcite, in this fracture and elsewhere in the fracture network[30]). There is thus strong evidence that the fungi were anaerobic, in a manner similar to fungi filtered from deep-water samples from fractured crystalline rocks in Finland[21]. The partly mineralized nature of the fungi and the degraded and matured carbon in the fungi speak against a modern origin. The only timing indication available is offered by the fluid inclusions in the calcite showing <50 °C, which rule out formation prior to the Mesozoic era based on the uplift history of the area[49, 50], but it should be emphasized that this is a maximum age estimate and not a direct age determination. In addition, the calcite crystals show fluid inclusion salinities that are much higher than in the present groundwater.

The $S^-$-isotope signatures indicate that the older pyrite generation precipitated from a more homogeneous fluid than the fine-grained younger generation of pyrite (Fig. 6). The relatively small variation in $\delta^{34}S$ of the older pyrite generation likely reflects formation during bacterial sulfate reduction (BSR) at relatively open system conditions. The younger fine-grained pyrite has a

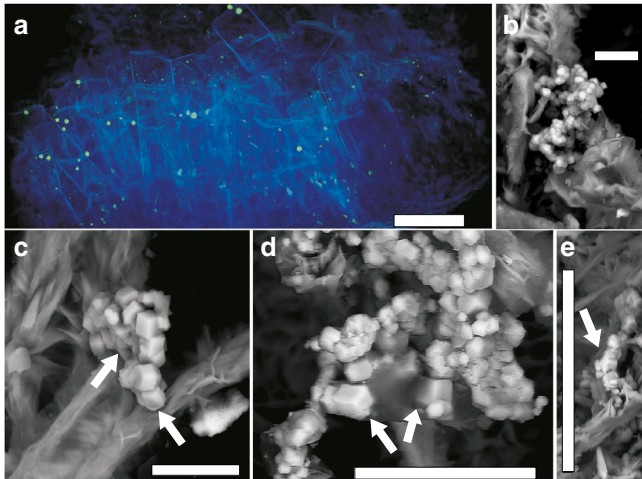

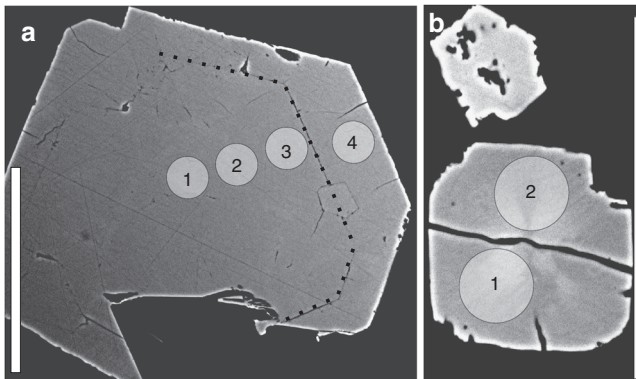

**Fig. 5** Characteristics of pyrite and spatial pyrite-hypha relation. (**a**) Synchrotron Radiation X-ray Tomographic Microscopy (SRXTM) volumetric rendering of zeolites, calcite and hyphae (*blue, transparent*) and pyrite (*green, opaque*). (**b–e**) Back-scattered Environmental Scanning Electron Microscopy (ESEM)-images in situ on the hyphal surface, (**b**) Fine-grained pyrite with hyphae (**c**) Two parallel hyphae of which the left one is older and has pyrite girdling around the outermost part due to SRB-related precipitation (*white arrow*), whereas the growth of the right hypha has been influenced by the already present pyrite crystals (*white arrow* highlights hypha by-passing pyrite). (**d, e**) Spatial relation of contemporaneous pyrite and hypha, showing that the pyrite has grown on the hypha, yet is in parts enclosed by it (*arrows*). The features in **b–e** indicate that SRB-related precipitation of pyrite was contemporaneous with fungal growth. Scale bars (**a**) 300 µm, (**b, c**) 10 µm, (**d**) 25 µm, (**e**) 50 µm

more clear relation to the hyphae than the older pyrite, and, hence, the discussion about the hyphae-SRB relation only takes the younger of these two pyrite generations into account. These fine-grained pyrite crystals were produced via the activity of SRB, because the low $\delta^{34}S$ values,are diagnostic for microbial transformation of sulfate to sulfide as $^{32}S_{SO4}$ is favoured over $^{34}S_{SO4}$ in the SRB metabolism[51]. The large span in $\delta^{34}S$ of this pyrite generation is interpreted as the result of Rayleigh type distillation where the $\delta^{34}S$ composition of the sulfate and thus of the precipitated pyrite became progressively higher as the sulfate pool was exhausted during BSR under closed system conditions[51]. Coexistence of the SRB producing these pyrite crystals and the fungi is thus possible and supported by a number of features. First, pyrite grows on original hyphal walls and not on parts that have been exposed by later breakdown/degradation caused by the core drilling or sample preparation, and clusters of pyrite enfold hyphae, forming almost a girdle-like structure that follows the hyphal morphology tightly (Fig. 5b, c); second, lack of hyphal degradation at the direct contact with the pyrite and no sign of pervasive replacement of hypha by pyrite, which is the common case in complete pyritization of fossils caused by heterotrophic bacterial activity[52, 53], argue against a situation where SRB only scavenged the fungal biomass; third, hyphal growth has been influenced by the presence of pyrite crystals; for example, hyphae have grown around existing pyrite that sits upon other hyphae (Fig. 5c) and pyrite crystals are partly enclosed by the hyphae (Fig. 5d, e); and fourth, spatial relation between pyrite, fatty acids and sugar compounds that resemble chitin, as revealed by Tof-SIMS analyses.

Anaerobic fungal species have no mitochondria and are unable to produce energy by either aerobic or anaerobic respiration[54, 55]. Instead, anaerobic fungi have hydrogenosomes, and produce mainly $H_2$, but also formate, lactate, acetate and carbon dioxide,

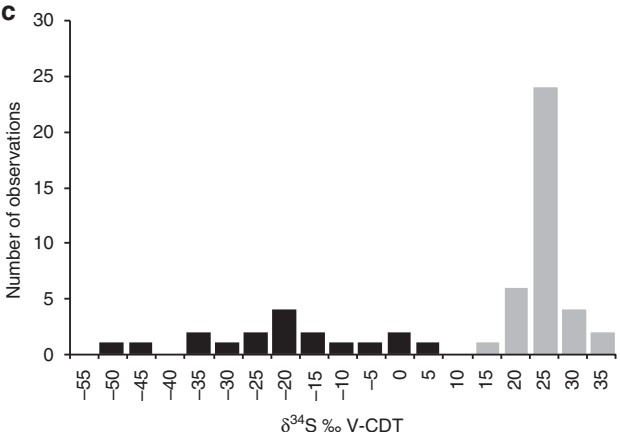

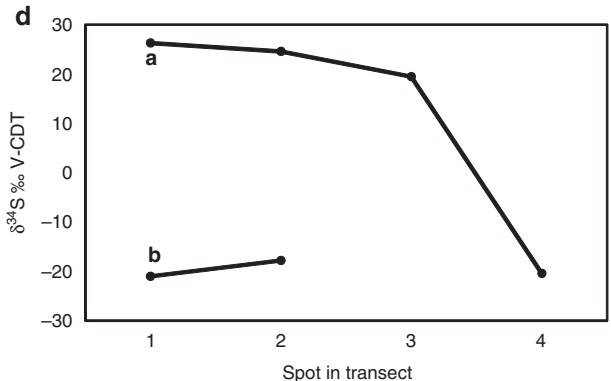

**Fig. 6** S isotope characteristics of pyrite. (**a**) Examples of polished cross-section of pyrite in a back-scattered Environmental Scanning Electron Microscopy (ESEM)-image. A boundary between the first and second generation of pyrite growth is visible and highlighted with a *stippled line*. *Circles* represent Secondary Ion Mass Spectrometry (SIMS)-spots (1–4), with values shown in **d**. (**b**) Polished cross-section of the young pyrite generation in a back-scattered ESEM-image. (**c**) Histogram of $\delta^{34}S$ values in pyrite. Two generations are distinguished (*black bars* = older, *grey bars* = younger). (**d**) $\delta^{34}S$ values from SIMS-transects in the crystals **a** and **b**. The younger generation, 4 in **a**, 1 and 2 in **b**, has significantly lower values than the older one (1–3 in **a**)[51]. Error bars (1 SD) are within the size of the symbols. Scale markers are 50 µm in **a** and **b**

as metabolic waste products[54, 56]. Anaerobic fungi consort with $H_2$-dependent methanogenic archaea in the rumen of ruminants, but potentially any $H_2$-dependent chemoautotrophic microorganism could be fuelled by anaerobic fungi in an anoxic environment[25], for instance SRB. Although the largest S-isotope fractionations observed in pure culture experiments have been associated with heterotrophic BSR[57], autotrophic BSR

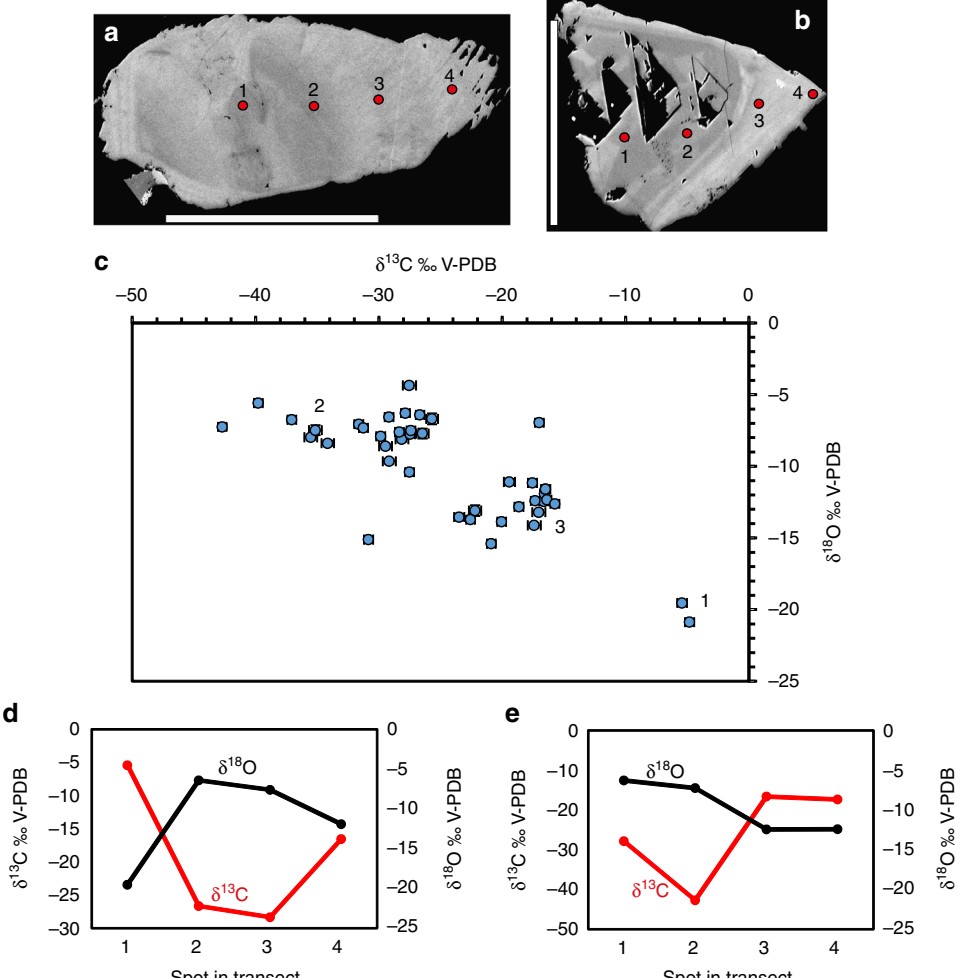

**Fig. 7** Stable isotope composition of calcite determined by Secondary Ion Mass Spectrometry (SIMS). (**a**, **b**) Examples of polished cross-sections of calcite crystals in high contrast back-scattered Environmental Scanning Electron Microscopy (ESEM)-images. Several different growth generations are visible (discerned by their back-scatter intensity, growth direction from left to right). Weathering is seen in the outermost (right) part of the crystal in **b**. Circles represent SIMS-spots (1–4). (**c**) All isotopic results (error bars, 1 SD, within size of symbols if not visible), including three different generations denoted according to relative age as seen in the polished cross-sections. (**d**, **e**) Transects in crystals shown in **a** and **b**, respectively. Error bars are within the size of the symbols. Scale bars are 500 μm (**b**) and 300 μm (**c**)

using $H_2$ also involves significant S-isotope fractionation ($\delta^{34}S_{H2S}$-$\delta^{34}S_{SO4}$ of up to 37‰), particularly at low $H_2$ concentrations and slow BSR rates[58]. Because the in situ rate of bacterial processes and generally also the concentrations of $H_2$ appear to be substantially lower in the granitic fractures[6, 59] than those manipulated in the laboratory, larger fractionation than reported from the laboratory appears reasonable under the extreme oligotrophy in the deep granite fractures. Hence, $H_2$ is a plausible electron donor for the SRB that produced the younger generation of pyrite, in line with the fact that the current groundwater at the site carries autotrophic microorganisms alongside heterotrophic ones[27]. We accordingly propose that $H_2$, and potentially some other substrate such as acetate, provided by anaerobic fungi, have triggered SRB growth (Fig. 8) and that, consequently, the intimate relationship between the fungal mycelium and the pyrite crystals represents a fossilized consortium of anaerobic fungi and SRB being the first record of these previously hypothesized communities[25]. This further suggests that the deep oligotrophic biosphere in crystalline continental rocks may be a neglected vast fungal habitat.

Hydrogen gas has been proposed to be an important substrate for deep subsurface lithoautotrophic ecosystems[60–63] and a limiting factor for the persistence of an indigenous SRB community[64, 65]. However, the formation and origin of $H_2$ remain elusive, and several different processes have been proposed, including radiolysis[60] over long time spans in the subsurface environment. Investigations from the Fennoscandian shield show highly variable $H_2$ concentrations in the deep groundwater (down to 1000 m depth) with concentrations up to 190 μl/l[27, 59], that are correlated with neither depth nor residence times of the waters, which in several cases are in the order of just a couple of thousand years[66]. Theseare certainly too short time periods for build-up of significant $H_2$ concentrations by radiolysis (cf. ref. [60]) implying that radiolysis is not the sole source of the elevated $H_2$ concentrations. Instead, based on our findings and ambiguous traces of fungi in the deep aquifer at Olkiluoto, Finland[21], we propose that subsurface fungi are neglected and likely significant providers of $H_2$ for autotrophic microbial processes in the oligotrophic crystalline continental crust.

Anaerobic fungi could potentially pose an environmental threat to barriers in geological repositories of toxic wastes, via two mechanisms: mediating direct bioerosion of the barrier system by chemical dissolution, as well as supporting an $H_2$-dependent SRB community capable of causing corrosion to copper canisters

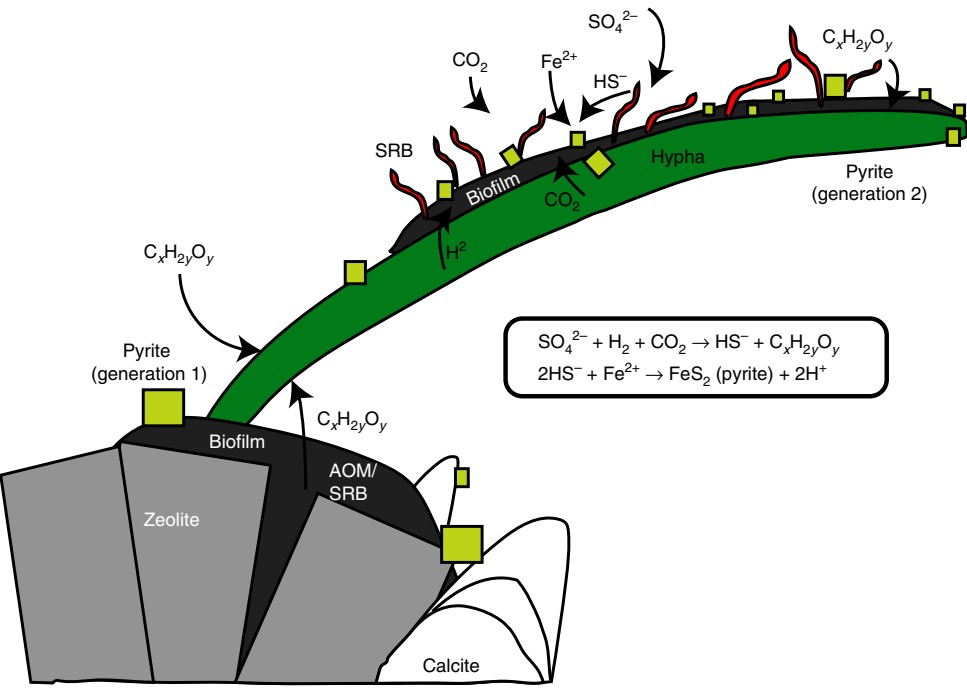

**Fig. 8** Conceptual model of coexistence and growth of fungi and sulfur reducing bacteria deep in crystalline bedrock. Growth of hypha starts at the biofilm on the zeolite surface, where sulfate reduction by sulfate-reducing bacteria (SRB) and anaerobic oxidation of methane (AOM) has led to oversaturation and precipitation of older calcite and pyrite. The biofilm provides organic carbon ($C_xH_{2y}O_y$) for the fungi to build biomass. The growing hypha metabolizes dissolved organic carbon both from the water and from the syntrophic SRB, which in turn thrive from the $H_2$ produced by the fungi. The SRB reduce sulfate ($SO_4^{2-}$) dissolved in the water autotrophically and produce $HS^-$ which reacts with dissolved $Fe^{2+}$ to form pyrite (now present on the hypha). The SRB use $CO_2$ from both the fungal metabolism and dissolved $CO_2$ in the water as carbon source

containing spent nuclear fuel[67]. Regarding the first mechanism, the extensive weathering of zeolites seen in the granite fracture in the present study, as well as similar observations in sub-seafloor basalts elsewhere[33, 34], calls for consideration when planning to use this group of minerals as geochemical barriers in subsurface storages. Zeolites have been planned to function as an ion-exchange retention barrier for the storage of high-level nuclear waste in the US[68, 69]. The cryptoendolithic behavior of anaerobic fungi may challenge the long-term stability of such systems, at least at low-temperatures. Regarding the second mechanism, our study shows a previously unseen relation between fungi and SRB at great depth in fractured granite which may, alongside long-term radiolytic $H_2$ consumption coupled to sulfate reduction, enhance sulfide levels in the deep groundwater aquifer. The ultimate result may be higher rates of sulfide-induced copper corrosion and CuS formation[64, 65]. The recognition of fungi in the subsurface realm, thus, indicates the presence of a previously neglected geobiological agent, the environmental impact and societal implications of which have yet to be accounted for.

## Methods

**SIMS.** Following sample characterisation in situ on the fracture surfaces using a Hitachi S-3400N scanning electron microscope (SEM) equipped with an integrated energy-dispersive spectroscopy (EDS) system under low-vacuum conditions, calcite and pyrite crystals were mounted in epoxy, polished to expose crystal cross-sections and examined again using SEM (to trace zonations). Intra-crystal SIMS-analysis of carbon, oxygen and sulfur isotopes were performed on a Cameca IMS1280 ion microprobe, at the NordSIM facility at the Museum of Natural History, Stockholm, Sweden. Analytical transects were made within the crystals. In total 47 analyses were made for $\delta^{13}C$ and 44 for $\delta^{18}O$ in calcite and 55 for $\delta^{34}S$ of pyrite in samples with AOM-signature in the calcite. Set up follows descriptions in ref. [30]. Sulfur was sputtered using a $^{133}Cs^+$ primary beam with 20 kV incident energy (10 kV primary, −10 kV secondary) and a primary beam current of ~ 1.5 nA, producing secondary ions from a slightly elliptical area of ~ 10 μm (long axis,

depth dimension is 1–2 μm). A normal incidence electron gun was used for charge compensation. Analyses were performed in automated sequences, with each analysis comprising a 70 s presputter to remove the gold coating over a rastered 25 × 25 μm area, centring of the secondary beam in the field aperture to correct for small variations in surface relief, and data acquisition in 16 4-s integration cycles. The magnetic field was locked at the beginning of the session using an NMR field sensor. Secondary ion signals for $^{32}S$ and $^{34}S$ were detected simultaneously using two Faraday detectors with a common mass resolution of 4860 ($M/\Delta M$). O was measured on two Faraday cages (FC) at mass resolution 2500, whereas C used a FC/EM combination, with mass resolution 2500 on the $^{12}C$ peak and 4000 on the $^{13}C$ peak to resolve it from $^{12}C^1H$. Data were normalized for instrumental mass fractionation using matrix matched standards which were mounted together with the sample mounts and analyzed after every sixth sample analysis. Isotope data from calcite were normalized using calcite standard S0161 and the Ruttan standard was used for pyrite (recommended values provided in Supplementary Tables 2 and 3). Precision was $\delta^{18}O$: ± 0.2‰, $\delta^{13}C$: ± 0.4‰ and $\delta^{34}S$: ± 0.13‰. Significant influence of organic carbon was avoided in the SIMS-analyses by careful spot placement to areas in the crystals without micro-fractures or inclusions, at a sufficient distance from grain-boundaries where fine-grained clusters of other minerals and remnants of organic material may appear. The uncertainty associated with potential organic inclusions and matrix composition is therefore considered to be insignificant compared to the isotopic variations.

**ToF-SIMS.** Right before ToF-SIMS analyses, the rock containing fractures with hyphae was cracked open, using clean tweezers (heptane, acetone and ethanol in that order), to expose fresh hyphae surfaces. The small pieces of rock containing the hyphae were then mounted with clean tweezers on double-sticky tape on a silica wafer. The ToF-SIMS analysis was performed on a ToF-SIMS IV (ION-TOF GmbH) by rastering a 25 keV $Bi_3^+$ beam (pulsed current of 0.1 pA) over an area of ~ 300 × 300 μm for 200–300 s. Analyses were performed in positive and negative mode at high mass resolution (bunched mode: $\Delta l \sim 3$ μm, $m/\Delta m \sim 2000$–4000 at $m/z$ 30). As a control, additional spectra were also acquired from the tape to confirm that samples had not been contaminated by it.

**SRXTM.** The tomographic measurements were carried out on the TOMCAT beamline at the Swiss Light Source, Paul Scherrer Institute, Villigen, Switzerland. A total of 1501 projections were acquired equiangularly over 180°, post-processed online and rearranged into flat- and darkfield-corrected sinograms. The beam

energy used for the six aliquots was 25 ($n = 1$), 30 (3), 35 (1) or 40 (1) keV, for maximum absorption contrast. Specimens were scanned with a 10x objective, resulting in cubic voxel dimensions of 0.65 µm. Visualization was done using Avizo 9.1.1 (FEI Company).

**GC–MS.** Extraction of fungi and a sea sand blank reference sample for analyses of organic compounds was carried out accordingly. Powder (1–2 g) was extracted with 4 ml of dichloromethane/methanol (3:1) in a Teflon-capped glass vial (ultrasonication, 35 min, 50 °C). The supernatant was decanted after centrifuging. Extraction was repeated with dichlormethane and afterwards with n-hexane. After evaporation of the combined extracts and re-dissolution in pure dichloromethane, the solvents were dried with $N_2$. Extracts were re-dissolved with 20 µl of n-Hexane and derivatized by adding 20 µl BSTFA/Pyridine and heated (40 °C, 1.5 h). Remnant powder was decarbonized with TMCS/Methanol (2:9) and derivatized by heating (80 °C, 1.5 h). The lipid fraction was separated by mixing with hexane and decanting the supernate. Extraction was repeated three times. The samples were dried with $N_2$, redissolved with 1 ml of n-Hexane and analyzed with GC/MS. For the analysis of the kerogen fraction ca 35 mg of sample extraction residues were mixed with sea sand (glowed for 2 h at 550 °C) and Molybdenum—catalyst. Catalytic hydropyrolysis (HyPy) was conducted with a constant $H_2$ flow at 5 l/min and a temperature program from 20 to 250 °C for 50 min and 250° to 500 °C for 8 min using a device from Strata Technology Ltd. (Nottingham, UK). The generated pyrolysate was absorbed on silica gel in the dry ice cooled trap tube. The HyPy pyrolysates were separated into an aliphatic, aromatic and polar fraction using column chromatography. To avoid any contamination, only pre-distilled solvents were used. All glassware used was first glowed at 500 °C. Solvent blank extracts (with pre-heated sea sand) were performed concomitantly as contamination controls and measured together with the investigated samples. One microlitre of each sample extract was analyzed with Thermo Trace 1310 GC coupled to a Thermo TSQ Quantum Ultra triple quadrupole MS. The GC was equipped with a fused silica capillary column (5 MS, 30 m lengths, 0.25 mm i.d., 0.1 µm film thickness, with He as carrier gas. The temperature program of the GC oven was 80 to 310 °C. The MS source was kept at 240 °C in electron impact mode at 25 eV ionization energy. Compounds and corresponding characteristic fragments detected with GC/MS are listed in Supplementary Table 1.

**Raman spectrometry.** Analysis was performed following[30] of fungi and for identification of minerals in a polished section using a Horiba instrument LabRAM HR 800 confocal laser Raman spectrometer equipped with a multichannel air-cooled CCD array detector. A low laser power 0.05 mW was used to avoid laser induced degradation of the sample. An Olympus BX41 microscope was coupled to the instrument. The laser beam was focused through a ×100 objective to obtain a spot size of about 1 µm. The accuracy of the instrument was controlled by repeated use of a silicon wafer calibration standard with a characteristic Raman line at 520.7 cm$^{-1}$. The Raman spectra were achieved with LabSpec 5 software.

**Fluid Inclusion Microthermometry.** Fluid inclusions were analyzed in handpicked calcite crystals (0.5–1.5 mm in size). Microthermometric analysesof fluid inclusions were made with a Linkam THM 600 stage mounted on a Nikon microscope utilizing a ×40 long working-distance objective. The working range of the stage is from −196° to +600 °C. The thermocouple readings were calibrated by means of SynFlinc synthetic fluid inclusions and well-defined natural inclusions in Alpine quartz. The reproducibility was ± 0.1 °C for temperatures below 40 and ± 0.5 °C for temperatures above 40 °C.

**Data availability.** All relevant data are included in the Supplementary material to this article.

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

## Acknowledgements

Thanks to the Swedish Nuclear Fuel and Waste Management Co. (SKB) and NOVA-FoU for giving access to drill cores from Laxemar. Thanks to L. Ilyinski and K. Lindén for assistance during SIMS-analysis and F. Marone for assistance at the TOMCAT beamline. This is NordSIM publication 511.

## Author contributions

H.D. initiated and planned the study together with M.I., carried out sampling, sample preparation, SEM- and SIMS-analyses, did the conceptual modelling and wrote the paper in collaboration with M.I. and M.E.Å. M.I. analyzed the fungi with SEM and processed SRXTM data together with S.B. and V.B. C.H. carried out biomarker analyses and interpretation, M.J.W. tuned the SIMS and reduced data, S.S. carried out the ToF-SIMS analyses. C.B. carried out fluid inclusion analyses and Raman spectroscopy (with M.I.). S.B. and V.B. analyzed the samples with SRXTM.

## Additional information

**Competing interests:** The authors declare no competing financial interests.

