## [Peer Review File · Nature Communications]

Reviewers' Comments:

Reviewer #1 (Remarks to the Author)

Please see my comments on the manuscript in the attached PDF document.

Reviewer #2 (Remarks to the Author)

The authors present an interesting study showing the potential co-existence of fungi and bacteria at -740m in granite. The paper is well written and the results interesting with implications for potential underground storage of nuclear waste. A range of analytical methods are used to back up the findings although most of the discussion focuses on the use of SIMS analysis of isotope ratios and electron microscopy.

Organic compounds are reportedly detected using GC-MS and ToF-SIMS.

It is surprising that the ToF-SIMS analysis seems to have been used to back up the GC analysis as opposed to providing unique information. Was imaging ToF-SIMS attempted and if so what did it show? It would be very interesting to see the transition between organically preserved and the mineralized hyphae in a ToF-SIMS image.

Also, where the listed chemicals the only species detected by the ToF-SIMS?

Bands related to organic/biological species were detected in the Raman analysis but the authors do not explain what species these represent or expand on the significance of these.

Minor comments, typos etc.:

Line 57. Ecosystem not ecosystems

Line 58. Insert "the" before continental

Line 126 and 130. Perhaps expand V-CDT and V-PDB.

Line 169. Should subseafloor be hyphenated?

Line 217. Insert "the" before sole.

Figure 3 axes need to be clearer.

Reviewer #3 (Remarks to the Author)

This manuscript reports a combination of SEM, SRTXM, GC-MS, Raman and isotope-sensitive mass spectrometry to characterize fossilised fungi from a deep mine rock sample, 740 m sub-surface.

The results are novel and of interest to the bio-geo-chemistry community. The proposal that an anaerobic fungi-bacteria consortium is involved is intriguing, and if true, would have implications for a number of deep sub-surface geochemical processes, as the authors discuss. While the evidence for fungi is clear, the only evidence for the presence of sulphate reducing bacteria (SRB) is S-isotope ratio data. I am not an expert on that method, so would ask the editor to consult with an expert to verify that the S-isotope ratio data is conclusive. Aside from the concern (which may simply reflect my ignorance in that area), I feel the paper is well constructed, the data appropriately presented, and the findings to be of broad interest and thus suitable for Nature Communications.

Some minor points to correct / clarify

Line 240 : H₂-dependent (spelling)

Line 297: "25-40 keV" Usually synchrotron X-ray tomography is performed with a monochromatised energy source. If that was the case here, what was the actual energy used ? If not, details of the measurements, or a reference to white beam tomography is needed. Was phase

or absorption contrast used ?

Reviewer #4 (Remarks to the Author)

Review of the manuscript

"Anaerobic consortia of fungi and sulfate reducing bacteria in deep granite fractures" by Henrik Drake with co-authors

The finding of fungi in continental crystalline rocks at great depth followed by the state-of-the-art analysis of the primary and secondary minerals surrounding fungal hyphae is the main strength of this work.

The topic is related to a very little studied area of geomicrobiology and could be of interest for Nature Communications readers.

The manuscript can be published only after thorough major revision.

There are following problems with the current version of the manuscript:

(1) The present way of the data presentation in some places are confusing probably due to the mistakes made in the legends and labels in the images and poor explanation of the illustrations in the Results section. For example Figs. 5 and 6 and the paragraph (lines 130-138) describing Fig. 6 are very difficult to follow: obviously the crystals on Fig.5 d) and e) were confused in the legend with c) and d) (see Lines 527-529). The lines 532-535 (the Fig. 6 legend) say: Weathering is clearly seen in the outermost part of crystal "b". It is unclear which image(s) and which features the authors referred to.

2) This manuscript presents some indirect evidence contributing to the previously suggested hypothesis about possible anaerobic fungi and SRB consortium in deep environments which is a great finding. But it should be remembered that it is not, in biological terms, a full direct evidence of such consortia. The manuscript would greatly benefit from the reinforcing your hypothesis with so called visual summary - a diagram/model explaining the possible relationships between fungi and SRB and their geochemical consequences in the light of your findings (e.g. visualizing Lines 197-235) clarifying it the for the diverse readers of Nature Communications. I would strongly suggest adding such model to the next version of a manuscript.

3) Line 187 - in biological/microbiological terms the use of the word "genetic" is incorrect.

4) Fig. 2: (i) I would put plural "hyphae" in Fig.2 a); (ii) black font is not visible in Fig.2 c) and d); (iii) Fig.2 f) is of poor resolution/quality, it is not clear and "crispy" enough for Nature Communications - it can be easily removed and be just explained in the text.

5) Fig. 3 is of poor quality. Is it possible to improve resolution, make it readable? It could be transferred to Supplementary material.

6) Fig.4 - black font is not visible, especially "Mineralized hypha" in b)

Line 154: The partly mineralized nature is in itself a rare finding. - How would you explain the partial biomineralization of fossilized hyphae? The heterogeneous biomineralization of hyphae has been previously observed in the experimental studies of rock weathering and uranium solids transformation by fungi under laboratory conditions and can be considered as quite common phenomenon for living mycelium (e.g. see Fomina et al., Environ. Microbiol., 2007; Geomicrobiol. J., 2010).

Reviewer #5 (Remarks to the Author)

Drake et al review

Anaerobic consortia of fungi and sulfate reducing bacteria in deep granite fractures

In this paper the authors report on the recently discovered mycelial networks found in Sweden at 740 meters depth in a drill core. They were found in a quartz vein in a partly open fracture. I have been asked to read the paper and comment on the sulfur isotope analysis done with SIMS on the pyrite found near the filaments. The primary findings are:

- There are two phases of pyrite growth, and the older one has a fairly narrow range of sulfur isotope composition (+23) while the more recent one has a wider range (over 50‰).
- There is a relationship between the pyrite and the filaments suggesting to the authors that they grew simultaneously.

I have no reason to doubt the evidence that these are fossilized fungi – and as a result I think the study has quite a large 'wow' factor. But I know very little about fungi. I suppose the crux of my reading of the story is the relationship between the fungi and the pyrite which suggests the relationship between the fungi and the sulfate reducing bacteria. I suppose I find this a bit weaker.

I am not overly convinced and can think of other scenarios that explain the isotopic data. The authors state that the fungi are anaerobic because the current fluids are anaerobic (Line 172-178), but if they were living in the Mesozoic and these are fossilized remains, then the fluids that are in the fracture network today seem to me to be a bit immaterial to whether it was anaerobic then. And they say that they are anaerobic because of the relationship with the pyrite, but I think that could be obtained with later stage reducing fluids remineralizing parts of the carbon in the matrix that remains.

And I suppose there is an issue with the order that these things come up. There are the statements on lines 172-178, but then it comes back Line 185-196 with perhaps slightly stronger arguments, but I think if the authors were to scan the literature of pyritization of fossils, they would find similar relationships and textures from the post depositional replacement of carbonaceous structures by microbial sulfate reduction and pyritization.

The sulfur isotope data is interesting but I don't know fully what to make of it. There is a first generation pyrite that is referred to once, but never returned to, that seems to have precipitated in completely closed system quantitative sulfate reduction where the isotope composition is high and constant. Then there are the secondary crystals that are smaller in shape and have a wider range of sulfur isotope composition. This wider range suggests they are supplied by diffusion (or some other fluid separating process) to the site of crystal growth but that doesn't require an insitu relationship between the fungi and microbial sulfate reduction. And I was troubled by the fact that microbial sulfate reduction that is feeding off the hydrogen produced by the fungi would have classically very low sulfur isotope fraction during sulfate reduction, which is inconsistent with the data which is very low.

So I am intrigued but I think that there is more thought that needs to happen, perhaps a better treatment of the literature, and a better presentation of the isotope data. Is the fossilized fungi enough to go on? I don't know. But I would take the pyrite textures and isotopes with a bit more of a grain of salt.

The isotope language is a bit colloquial. I am always getting slammed for this in review and so I appreciate the terminology, but 'heavier' and 'lighter' really should be replaced with 'enriched in the x isotope' or 'enriched in the y isotope'. Or higher or lower.

Figure 5 – remove 'it is clear' from figure caption. Also the last bit I think refers to iii in d) (not c). Also it would help if e was in two colors so you could distinguish the isotope results from what the authors interpret as one growth phase from what they interpret as the second growth phase.

Reviewer 1

I have been asked to specifically comment on the application of fluid inclusions and Raman spectroscopic analyses in the manuscript, so I will stick to the editor's request with my comments.

The manuscript focuses on fossil and active fungi discovered at depth in fractured continental crystalline rocks, and uses fluid inclusions to determine the formation temperature of fracturefilling calcite formed associated with the fungi, and Raman spectroscopy to identify some of the other mineral phases and organic compounds present in the fractures. Overall the manuscript is very interesting, well written, and the presented data and illustrations support most of the findings. Some parts, however, need to be modified, in order to fit the described findings, as I comment below.

The Raman spectroscopic results presented seem straight forward, with the probable standard procedure followed by the authors of acquiring Raman spectra on a molecular compound and compare the spectra with available spectral databases to identify the compounds. As no details are presented on the obtained spectra in the manuscript, nor in the supplementary material, I cannot further comment on the quality of the results. But having a limited space in the journal and knowing the focus and other results presented in the manuscript, further details are probably not necessary to support these observations.

Response: Indeed, Raman spectroscopy is not one of the main methods used. We have nevertheless added some more interpretations from the Raman observations, see reply to reviewer 2 below.

However, I have some issues regarding the fluid inclusion results, even if the role of this technique in interpreting observations also seems secondary to other methods presented. The authors describe in the methods section how they proceeded with the fluid inclusions microthermometry, but they do not present the data in any form, table or figure, so I cannot comment on the quality of the data. It is not clear, for example, whether the inclusions are primary or secondary in nature, e.i., trapped during the formation of host calcite, or after, along healed microfractures (see Goldstein, 2001, referenced in manuscript). The results mentioned in lines 136-138 are the only reference to any fluid inclusions observations, where the authors comment on salinity and trapping temperature. However, the formulation of the sentence is misleading, I think. In the present format the authors imply that the salinity of single phase inclusions indicates formation temperatures below 50 °C. However, and I sense that the authors know this as well (based on the reference they cite), the low trapping temperatures are indicated by the presence of single phase inclusions, not salinity. The sentence needs to be reformatted. Moreover, to reach this conclusion no microthermometry is necessary, as the presence of consistently single phase aqueous inclusions within a fluid inclusion assemblage readily indicate low temperatures, usually <50 °C (as described in their reference to Goldstein (2001)).

Response: The fluid inclusion results are now presented in a more detailed text at lines 145-149 and in a figure (Supplementary Fig. 3), describing and showing the primary nature of the inclusions and the microthermometrical results/interpretations in greater detail. We have updated the sentence formerly on lines 136-138 accordingly: "Single phase liquid fluid inclusions in the calcite (Supplementary Fig. 3) indicate formation temperatures below 50°C³². The inclusions show low eutectic melting temperature close to -50°C which points to a

composition where CaCl_2 is the most abundant salt. Final ice melting temperatures in the range -22.4° to -23.5°C correspond to salinities of 21.7-22.2 wt. % CaCl_2 ”, (lines 145-149).

The methods section raises one more concern. The authors describe that in order to determine the homogenization temperature of the single phase inclusion, a vapor bubble (described as “gas”, incorrectly) needs to be produced by heating the sample to 150°C . Indeed, in many cases this produces a vapor bubble by stretching the fluid inclusion. However, the temperature obtained following the sample reheating would result in a temperature that is not necessarily the true homogenization temperature of the fluid inclusion. Heating of a fluid phase in a container with fixed volume (the inclusion) will lead to the expansion of the fluid volume. Since the volume of the container is fixed, the fluid will apply a pressure on the walls of the container. This pressure could force the container, e.i. the inclusion in calcite, to expand, or stretch, resulting in a larger volume, filled by a vapor phase. But since this new inclusion volume is (unknowingly) larger than the original inclusion, the homogenization temperature of the newly formed two-phase inclusion will be (unknowingly) higher than the true homogenization temperature of the original inclusion. To avoid this issue, single phase inclusions ought to be cooled, without freezing the inclusions, or use other techniques to induce a vapor bubble (e.g., with femtosecond laser; Krüger et al., 2007), to prevent inclusion stretching. But without seeing the data and the observed temperature ranges, I cannot further comment on this issue.

Reference:

Krüger, Y., Stoller, P., Rička, J., and Frenz, M., 2007. Femtosecond lasers in fluid inclusion analysis: Overcoming metastable phase states: *European Journal of Mineralogy*, v. 19, p. 693–706, doi:10.1127/0935-1221/2007/0019-1762

Response: We agree that the text was unclear. Therefore, we have updated the text accordingly (lines 382-387): “All analyzed fluid inclusions were primary in nature and of single phase (liquid) type. The salinities of the inclusions were calculated from the final ice melting temperature, but to avoid metastable ice melting and incorrect temperatures, the single-phase all-liquid inclusions were heated to 150°C , before freezing, in order to stretch them and to generate an artificial vapour bubble.”

Reviewer 2

The authors present an interesting study showing the potential co-existence of fungi and bacteria at -740m in granite. The paper is well written and the results interesting with implications for potential underground storage of nuclear waste. A range of analytical methods are used to back up the findings although most of the discussion focuses on the use of SIMS analysis of isotope ratios and electron microscopy.

Organic compounds are reportedly detected using GC-MS and ToF-SIMS.

It is surprising that the ToF-SIMS analysis seems to have been used to back up the GC analysis as opposed to providing unique information. Was imaging ToF-SIMS attempted and if so what did it show? It would be very interesting to see the transition between organically preserved and the mineralized hyphae in a ToF-SIMS image.

Also, where the listed chemicals the only species detected by the ToF-SIMS?

Response: A ToF-SIMS figure of partly mineralized hyphae was added (Supplementary Figure 2), as well as text referring to and describing this figure and the results in the main text (lines 107-11: “The Time-of-Flight (ToF)-SIMS results of partly mineralized hyphae show possible association of sulfur (pyrite), SO_4H^- and PO_2^- with sugar fragments (potentially chitin-related fragment as the $\text{C}_3\text{H}_3\text{O}_2^-$ ion produces a strong signal in chitin standard spectrum, but also in spectra of other types of sugar molecules), and fatty acids ($\text{C}_{16:0}$ and $\text{C}_{18:1}$ at m/z 255.23 and 281.25, respectively, Supplementary Fig. 2).”.

We could detect the transition from preserved to mineralized hyphae in high resolution SEM, where we could investigate large areas of mycelium, but in the ToF-SIMS we had to extract smaller fragments of hyphae and it was not straight-forward to separate the partially mineralized hyphae during selection of single specimens to put into the ToF-SIMS chamber. Transition phase hyphae were mostly present deep within the mycelium and we choose not to destroy the whole mycelium for the ToF-SIMS investigations because the intact mycelium was to be used in SRXTM afterwards. Therefore, the ToF-SIMS measurement was not made on the most representative transition between mineralized and organically preserved hyphae. ToF-SIMS was not used to back up the GC analysis, and the Supplementary Table 1 now only shows compounds detected using GC-MS, and the ToF-SIMS observations are listed in Supplementary Figure 2.

Bands related to organic/biological species were detected in the Raman analysis but the authors do not explain what species these represent or expand on the significance of these.

Response: We have added some more information about this on lines 97-102: “Raman spectroscopic analysis on the mineralized filaments and the carbonaceous filaments shows characteristics of typical organic compounds with three clear peaks at 1300, 1440 and 1660 cm^{-1} which may be attributed to CH_2 - and CH_3 deformations and $\text{C}=\text{C}$ stretching vibrations similar to what has been found in fatty acids³¹. The spectra further show several bands between 2720 and 3300 cm^{-1} with the most intense at 2850 and 2930 cm^{-1} assigned to CH_2 and CH_3 stretching vibrations.”

Minor comments, typos etc.:

Line 57. Ecosystem not ecosystems

Line 58. Insert “the” before continental

Line 126 and 130. Perhaps expand V-CDT and V-PDB.
Line 169. Should subseafloor be hyphenated?
Line 217. Insert “the” before sole.
Figure 3 axes need to be clearer.

Response: We have followed all of these advices.

Reviewer 3

This manuscript reports a combination of SEM, SRTXM, GC-MS, Raman and isotope-sensitive mass spectrometry to characterize fossilised fungi from a deep mine rock sample, 740 m sub-surface.

The results are novel and of interest to the bio-geo-chemistry community. The proposal that an anaerobic fungi-bacteria consortium is involved is intriguing, and if true, would have implications for a number of deep sub-surface geochemical processes, as the authors discuss. While the evidence for fungi is clear, the only evidence for the presence of sulphate reducing bacteria (SRB) is S-isotope ratio data. I am not an expert on that method, so would ask the editor to consult with an expert to verify that the S-isotope ratio data is conclusive. Aside from the concern (which may simply reflect my ignorance in that area), I feel the paper is well constructed, the data appropriately presented, and the findings to be of broad interest and thus suitable for Nature Communications.

Response: See detailed reply and updates in response to reviewer 5.

Some minor points to correct / clarify

Line 240 : H₂-dependent (spelling)

Response: Spelling has been corrected.

Line 297: “25-40 keV” Usually synchrotron X-ray tomography is performed with a monochromatised energy source. If that was the case here, what was the actual energy used? If not, details of the measurements, or a reference to white beam tomography is needed. Was phase or absorption contrast used?

Response: The energy was different for the six aliquots scanned, and absorption contrast was used. The text has been updated to explain this: “The beam energy used for the six aliquots was 25 (n=1), 30 (3), 35 (1) or 40 (1) keV, for maximum absorption contrast.”

Reviewer 4

The finding of fungi in continental crystalline rocks at great depth followed by the state-of-the-art analysis of the primary and secondary minerals surrounding fungal hyphae is the main strength of this work.

The topic is related to a very little studied area of geomicrobiology and could be of interest for Nature Communications readers.

The manuscript can be published only after thorough major revision.

There are following problems with the current version of the manuscript:

(1) The present way of the data presentation in some places are confusing probably due to the mistakes made in the legends and labels in the images and poor explanation of the illustrations in the Results section. For example Figs. 5 and 6 and the paragraph (lines 130-138) describing Fig. 6 are very difficult to follow: obviously the crystals on Fig.5 d) and e) were confused in the legend with c) and d) (see Lines 527-529). The lines 532-535 (the Fig. 6 legend) say: Weathering is clearly seen in the outermost part of crystal “b”. It is unclear which image(s) and which features the authors referred to.

Response: We thank the reviewer for noticing and highlighting these unclaritys. We have accordingly updated both the text and caption to make it clearer that we describe different precipitation events, e.g. on lines (139-142): “The calcite crystals have distinct growth zonation that represent at least three precipitation events (generations) with specific stable isotope compositions. Analytical transects within the crystals show that the oldest generation has highest $\delta^{13}\text{C}$ and lowest $\delta^{18}\text{O}$ (Fig. 7a,d, zoned crystal with growth direction from left to right).”. Yes, there was a labelling-error in figure 5g. We have corrected it and divided the figure into two figures for clarity, and updated and clarified the caption.

2) This manuscript presents some indirect evidence contributing to the previously suggested hypothesis about possible anaerobic fungi and SRB consortium in deep environments which is a great finding. But it should be remembered that it is not, in biological terms, a full direct evidence of such consortia. The manuscript would greatly benefit from the reinforcing your hypothesis with so called visual summary - a diagram/model explaining the possible relationships between fungi and SRB and their geochemical consequences in the light of your findings (e.g. visualizing Lines 197-235) clarifying it for the diverse readers of Nature Communications. I would strongly suggest adding such model to the next version of a manuscript.

Response: We have now further emphasized and explained that the evidence of such consortia is based on observations from a fossilized system (lines 256-258: “...the intimate relationship between the fungal mycelium and the pyrite crystals represents a fossilized consortium of anaerobic fungi and SRB...”). We also thank the reviewer for the good suggestion of making a visual summary. We have accordingly made such a summary and included it in the manuscript as Figure 8.

3) Line 187 – in biological/microbiological terms the use of the word “genetic” is incorrect.

Response: “Genetic” was changed to “co-genetic”, in order to avoid such confusion.

4) Fig. 2: (i) I would put plural “hyphae” in Fig.2 a); (ii) black font is not visible in Fig.2 c)

and d); (iii) Fig.2 f) is of poor resolution/quality, it is not clear and “crispy” enough for Nature Communications – it can be easily removed and be just explained in the text.

Response: We have updated the figures thoroughly in accordance with these constructive suggestions to improve the presentation. Regarding figure 2f, we have cropped the image significantly, leaving out the part that was out of focus.

5) Fig. 3 is of poor quality. Is it possible to improve resolution, make it readable? It could be transferred to Supplementary material.

Response: We have increased resolution and readability.

6) Fig.4 – black font is not visible, especially “Mineralized hypha” in b)

Response: We have updated the image to make it readable.

Line 154: The partly mineralized nature is in itself a rare finding. – How would you explain the partial biomineralization of fossilized hyphae? The heterogeneous biomineralization of hyphae has been previously observed in the experimental studies of rock weathering and uranium solids transformation by fungi under laboratory conditions and can be considered as quite common phenomenon for living mycelium (e.g. see Fomina et al., *Environ. Microbiol.*, 2007; *Geomicrobiol. J.*, 2010).

Response: We have added the references, and also added a short description of that this phenomenon has been observed in the laboratory and that it is a particularly rare finding for subsurface environments. More details on the mineralization of the hyphae are included on lines 170-182.

Reviewer 5

Anaerobic consortia of fungi and sulfate reducing bacteria in deep granite fractures

In this paper the authors report on the recently discovered mycelial networks found in Sweden at 740 meters depth in a drill core. They were found in a quartz vein in a partly open fracture. I have been asked to read the paper and comment on the sulfur isotope analysis done with SIMS on the pyrite found near the filaments. The primary findings are:

- There are two phases of pyrite growth, and the older one has a fairly narrow range of sulfur isotope composition (+23) while the more recent one has a wider range (over 50‰).
- There is a relationship between the pyrite and the filaments suggesting to the authors that they grew simultaneously.

I have no reason to doubt the evidence that these are fossilized fungi – and as a result I think the study has quite a large ‘wow’ factor. But I know very little about fungi. I suppose the crux of my reading of the story is the relationship between the fungi and the pyrite which suggests the relationship between the fungi and the sulfate reducing bacteria. I suppose I find this a bit weaker.

I am not overly convinced and can think of other scenarios that explain the isotopic data. The authors state that the fungi are anaerobic because the current fluids are anaerobic (Line 172-178), but if they were living in the Mesozoic and these are fossilized remains, then the fluids that are in the fracture network today seem to me to be a bit immaterial to whether it was anaerobic then. And they say that they are anaerobic because of the relationship with the pyrite, but I think that could be obtained with later stage reducing fluids remineralizing parts of the carbon in the matrix that remains.

Response: In the updated version, the arguments regarding anoxic conditions have been strengthened by more descriptions, in particular the addition of more lines of evidence for prolonged anoxic conditions on lines 199-208: “1) there is pyrite in relation to the hyphae, 2) oxidation-related alteration features are not detected on any of the pyrite generations in the fractures, in contrast to pyrite at shallower depth where waters with dissolved oxygen have infiltrated⁴⁶, 3) Ce(IV) and positive Ce anomalies, which are indicative of oxidising conditions, are frequent in fracture coatings in the upper 10 m of the bedrock but absent below that depth⁴⁸, and, 4) abundant signs of anaerobic oxidation of methane (¹³C-depleted calcite, in this fracture and elsewhere in the fracture network³⁰), are proxies for prolonged anoxic conditions in the deep groundwater aquifer. There is thus strong evidence that the fungi were anaerobic, in a manner similar to fungi filtered from deep-water samples from fractured crystalline rocks in Finland²¹.”. Also, we have not concluded that the hyphae are Mesozoic in age; merely that this reflects their theoretical maximum age. We have clarified this to avoid confusion regarding the timing estimates, as follows “The only timing indication available is offered by the fluid inclusions in the calcite showing <50°C, which rule out formation prior to the Mesozoic era based on the uplift history of the area^{49,50}, but it should be emphasized that this is a maximum age estimate and not a direct age determination.” (lines 209-212).

And I suppose there is an issue with the order that these things come up. There are the statements on lines 172-178, but then it comes back Line 185-196 with perhaps slightly stronger arguments, but I think if the authors were to scan the literature of pyritization of

fossils, they would find similar relationships and textures from the post depositional replacement of carbonaceous structures by microbial sulfate reduction and pyritization.

Response: In the revised version, we have provided more SEM-images of the pyrite-fungi relationships that strengthen and clarify the co-genetic relationship (Fig. 5, with updated and more detailed corresponding text in the captions and main text) on lines 226-238, accordingly: “Coexistence of the SRB producing these pyrite crystals and the fungi are thus possible, and supported by a number of features: 1) Pyrite grows on original hyphal walls and not on parts that have been exposed by later breakdown/degradation caused by the core drilling or sample preparation, and clusters of pyrite enfold hyphae, forming almost a girdle-like structure that follows the hyphal morphology tightly (Fig. 5b, c). 2) Lack of hyphal degradation at the direct contact with the pyrite, and no sign of pervasive replacement of hypha by pyrite that is the common case in complete pyritization of fossils caused by heterotrophic bacterial activity^{52,53}, speak against a situation where SRB only scavenged the fungal biomass. 3) Hyphal growth has been influenced by the presence of pyrite crystals, for example, hyphae have grown around existing pyrite that sits upon other hyphae (Fig. 5c) and pyrite crystals are partly enclosed by the hyphae (Fig. 5d, e). 4) Spatial relation between pyrite, fatty acids and sugar compounds that resemble chitin, as revealed by ToF-SIMS analyzes.” This includes both a discussion regarding differences of our findings to the complete pyritization of fossils found elsewhere (our findings point to a co-genetic growth and not a complete occupation of the fossil structure) and an additional linkage to the ToF-SIMS results.

The sulfur isotope data is interesting but I don't know fully what to make of it. There is a first generation pyrite that is referred to once, but never returned to, that seems to have precipitated in completely closed system quantitative sulfate reduction where the isotope composition is high and constant. Then there are the secondary crystals that are smaller in shape and have a wider range of sulfur isotope composition. This wider range suggests they are supplied by diffusion (or some other fluid separating process) to the site of crystal growth but that doesn't require an insitu relationship between the fungi and microbial sulfate reduction. And I was troubled by the fact that microbial sulfate reduction that is feeding off the hydrogen produced by the fungi would have classically very low sulfur isotope fraction during sulfate reduction, which is inconsistent with the data which is very low.

Response: The S-isotope composition of the pyrite is indeed SRB-related which confirms that the pyrites were formed by bacterial sulfate reduction, which was the purpose of the S-isotope measurements. The syntrophic SRB-fungi relationship interpretation is in turn based on the microscope observations (mostly ESEM, and SRXTM). The older pyrites are not as obviously associated with the hyphae as the younger pyrite generation and therefore we focus on the younger pyrites in our interpretations (some more details about the older pyrite are now given in the text). Experiments have shown that fractionation during H₂-based autotrophic BSR can be substantial (up to 37 per mil has been evidenced by Hoek et al., 2006, *Geochim. Cosmochim. Acta*), particularly at low H₂ concentrations and slow BSR rates, which we have argued for in the updated version. The rates of microbial processes in this environment are indeed very slow, particularly compared to those manipulated in laboratory experiments. Therefore, it is not unreasonable to believe that even larger fractionation than 37 per mil can occur in this environment. However, we cannot completely exclude heterotrophic BSR as an explanation for the absolutely lowest $\delta^{34}\text{S}_{\text{pyrite}}$ detected. Nevertheless, a heterotrophic BSR process in addition to the autotrophic BSR is not unlikely in the deep groundwater aquifer, as microbiological investigations in this area have confirmed co-

occurring heterotrophic and autotrophic microorganisms in most of the deep water conducting fractures (Hallbeck and Pedersen, 2008, *Appl Geochem*). We have therefore strengthened the discussion at lines 214-226 and 239-258, accordingly:

“The S-isotope signatures indicate that the oldest pyrite generation precipitated from a more homogeneous fluid than the fine-grained younger generation of pyrite (Fig. 6). The relatively small variation in $\delta^{34}\text{S}$ of the older pyrite generation likely reflects formation during bacterial sulfate reduction (BSR) at relatively open system conditions. The younger fine-grained pyrite has a more clear relation to the hyphae than the older pyrite has, and hence the discussion about the hyphae-SRB relation only takes the younger of these two pyrite generations into account. These fine-grained pyrite crystals were produced via the activity of sulfate reducing bacteria (SRB), because the low $\delta^{34}\text{S}$ values are diagnostic for microbial transformation of sulfate to sulfide as $^{32}\text{S}_{\text{SO}_4}$ is favoured over $^{34}\text{S}_{\text{SO}_4}$ in the SRB metabolism⁵¹. The large span in $\delta^{34}\text{S}$ of this pyrite generation is interpreted as the result of Rayleigh type distillation where the $\delta^{34}\text{S}$ composition of the sulfate and thus of the precipitated pyrite has been progressively higher as the sulfate pool has been exhausted during BSR under closed system conditions⁵¹. “

“Anaerobic fungal species have no mitochondria and are unable to produce energy by either aerobic or anaerobic respiration^{54,55}. Instead, anaerobic fungi have hydrogenosomes, and produce mainly H_2 , but also formate, lactate, acetate and carbon dioxide, as metabolic waste products^{54,56}. Anaerobic fungi consort with H_2 -dependent methanogenic archaea in the rumen of ruminants, but potentially any H_2 -dependent chemoautotrophic microorganism could be fuelled by anaerobic fungi in an anoxic environment²⁵, for instance SRB. Although the largest S-isotope fractionations observed in pure culture experiments have been associated with heterotrophic BSR⁵⁷, autotrophic BSR using H_2 also involve significant S-isotope fractionation ($\delta^{34}\text{S}_{\text{H}_2\text{S}} - \delta^{34}\text{S}_{\text{SO}_4}$ of up to 37‰), particularly at low H_2 concentrations and slow BSR rates⁵⁸. Because the *in situ* rate of bacterial processes and generally also the concentrations of H_2 appear to be substantially lower in the granitic fractures^{6,59} than those manipulated in the laboratory, larger fractionation than reported from the laboratory appear reasonable under the extreme oligotrophy in the deep granite fractures. Hence, H_2 is a fully possible electron donor for the SRB that produced the younger generation of pyrite, in line with the fact that the current groundwater at the site carry autotrophic microorganisms alongside heterotrophic ones²⁷. We accordingly propose that H_2 , and potentially some other substrate such as acetate, provided by anaerobic fungi have triggered SRB growth (Fig. 8) and that, consequently, the intimate relationship between the fungal mycelium and the pyrite crystals represents a fossilized consortium of anaerobic fungi and SRB being the first record of these previously hypothesized communities²⁵.”

So I am intrigued but I think that there is more thought that needs to happen, perhaps a better treatment of the literature, and a better presentation of the isotope data. Is the fossilized fungi enough to go on? I don't know. But I would take the pyrite textures and isotopes with a bit more of a grain of salt.

Response: As described above, we have added some more examples of the pyrite-hypha-relation and discussed/compared them in a broader context, as well as including a more thorough interpretation of the S-isotope data.

The isotope language is a bit colloquial. I am always getting slammed for this in review and so I appreciate the terminology, but ‘heavier’ and ‘lighter’ really should be replaced with

‘enriched in the x isotope’ or ‘enriched in the y isotope’. Or higher or lower.

Response: We have changed to higher and lower, which we prefer, but many reviewers do not agree.

Figure 5 – remove ‘it is clear’ from figure caption. Also the last bit I think refers to iiiii in d) (not c). Also it would help if e was in two colors so you could distinguish the isotope results from what the authors interpret as one growth phase from what they interpret as the second growth phase.

Response: Figure 5 has been split into two figs (5 and 6) and color coding has been added to the histogram. The reference in one of the subfigures to two of the other subfigures has been corrected.

Reviewers' Comments:

Reviewer #1:

Remarks to the Author:

I have been asked to specifically comment on the application of fluid inclusions and Raman spectroscopic analyses while originally reviewed this manuscript. I think the authors addressed all of my (and the other reviewers') questions during their revision of the manuscript, and by adjusting the text and adding extra figures they made the manuscript clearer, with better support for their findings. I think the manuscript is ready for publishing in Nature Communications, and support its acceptance for publication in the present format.

Reviewer #2:

Remarks to the Author:

The authors appear to have adequately addressed all of my previous comments regarding this manuscript.

Reviewer #3:

Remarks to the Author:

The revised manuscript and associated responses adequately address the concerns raised by the reviewers in the areas on my expertise. Accept for publication

Reviewer #4:

Remarks to the Author:

The points raised by me in the previous round of review have been satisfactorily addressed by the authors. I am very pleased with the addition of the Conceptual model of fungal-SRB coexistence and growth deep in crystalline bedrock. This version of the manuscript can be published in Nature Communication.

Reviewer #5:

Remarks to the Author:

I have reread the paper and the response to my concerns. I think the arguments have been greatly strengthened in response to my comments, and I thank the authors for this. I am particularly grateful that they discussed the open system and closed system and sulfur isotope fractionation in more detail with respect to the growth phases of pyrite.

I think the paper is ready for publication. In some of the newly written bits the language needs some editing but I am happy with the paper progressing to publication.

Response to reviewers' comments. Manuscript NCOMMS-16-29994, “Anaerobic consortia of fungi and sulfate reducing bacteria in deep granite fractures”

We acknowledge the positive assessment provided by the five reviewers. The reviewer comments are listed below, and as can be seen there are no new queries except that the language should be checked in the new text parts (which has now been done, and the resulting minor changes can be traced in the manuscript file with track changes). Therefore we have not included any detailed responses below.

Reviewer assessments:

Reviewer #1 (Remarks to the Author):

I have been asked to specifically comment on the application of fluid inclusions and Raman spectroscopic analyses while originally reviewed this manuscript. I think the authors addressed all of my (and the other reviewers') questions during their revision of the manuscript, and by adjusting the text and adding extra figures they made the manuscript clearer, with better support for their findings. I think the manuscript is ready for publishing in Nature Communications, and support its acceptance for publication in the present format.

Reviewer #2 (Remarks to the Author):

The authors appear to have adequately addressed all of my previous comments regarding this manuscript.

Reviewer #3 (Remarks to the Author):

The revised manuscript and associated responses adequately address the concerns raised by the reviewers in the areas on my expertise. Accept for publication

Reviewer #4 (Remarks to the Author):

The points raised by me in the previous round of review have been satisfactorily addressed by the authors. I am very pleased with the addition of the Conceptual model of fungal-SRB coexistence and growth deep in crystalline bedrock. This version of the manuscript can be published in Nature Communication.

Reviewer #5 (Remarks to the Author):

I have reread the paper and the response to my concerns. I think the arguments have been greatly strengthened in response to my comments, and I thank the authors for this. I am particularly grateful that they discussed the open system and closed system and sulfur isotope fractionation in more detail with respect to the growth phases of pyrite.

I think the paper is ready for publication. In some of the newly written bits the language needs some editing but I am happy with the paper progressing to publication.